# Factors associated with delayed entry into HIV medical care after HIV diagnosis in a resource-limited setting: Data from a cohort study in India

Gerardo Alvarez-Uria

Department of Infectious Diseases, Rural Development Trust Hospital, Bathalapalli, AP, India

## ABSTRACT

Studies from sub-Saharan Africa have shown that a substantial proportion of patients diagnosed with HIV enter into HIV medical care late. However, data from low or middle-income countries outside Africa are scarce. In this study, we investigated risk factors associated with delayed entry into care stratified by gender in a large cohort study in India. 7701 patients were diagnosed with HIV and 5410 entered into care within three months of HIV diagnosis. Nearly 80% entered into care within a year, but most patients who did not enter into care within a year remained lost to follow up or died. Patient with risk factors related to having a low socio-economic status (poverty, being homeless, belonging to a disadvantaged community and illiteracy) were more likely to enter into care late. In addition, male gender and being asymptomatic at the moment of HIV infection were factors associated with delayed entry into care. Substantial gender differences were found. Younger age was found to be associated with delayed entry in men, but not in women. Widows and unmarried men were more likely to enter into care within three months. Women belonging to disadvantaged communities or living far from a town were more likely to enter into care late. The results of this study highlight the need to improve the linkage between HIV diagnosis and HIV treatment in India. HIV programmes should monitor patients diagnosed with HIV until they engage in HIV medical care, especially those at increased risk of attrition.

## INTRODUCTION

There are approximately 34 million people living with HIV worldwide (*UNAIDS, 2012*). Despite the impressive roll-out of antiretroviral therapy (ART) programmes worldwide, including in low and middle-income countries, 1.7 million people died of HIV-related pathologies in 2011 (*UNAIDS, 2012*).

People who are diagnosed with HIV are linked to healthcare facilities able to provide ART and, ideally, ART should be started before the development of opportunistic

Corresponding author
Gerardo Alvarez-Uria,
gerardouria@gmail.com

infections. However, the majority of patients enter into care late both in developed and developing countries (*Adler, Mounier-Jack & Coker, 2009*; *Althoff et al., 2010*; *Alvarez-Uria et al., 2012c*; *Girardi, Sabin & Monforte, 2007*). One of the most important reasons for this late presentation is the poor linkage between healthcare centres performing HIV testing and ART centres (*Bassett et al., 2010*; *Kranzer et al., 2010*; *Larson et al., 2010*; *Losina et al., 2010*). In ART centres, the measurement of the CD4 lymphocyte count is the first step of the assessment of HIV infected patients. Studies from sub-Saharan Africa have shown that only 72% (95% confidence interval [CI], 60–84%) of patients diagnosed with HIV have a CD4 count measured (*Mugglin et al., 2012*), and between one-third and two-thirds provide samples for CD4 counts within 2–3 months of the HIV positive test (*Rosen & Fox, 2011*). However, data from resource-limited settings outside Africa are scarce.

With 2.4 million HIV infected people, India has the third largest burden of HIV worldwide (*National AIDS Control Organisation, 2011*), and two thirds of the patients live in rural areas (*UNAIDS/WHO, 2010*). In this study, we describe the engagement in care of people diagnosed with HIV in a large HIV cohort study in India. In particular, we aimed to find risk factors associated with delayed entry into HIV care in an Indian rural setting.

## METHODS

### Setting

The study was performed in Anantapur, a district situated in Andhra Pradesh, India. In Anantapur, 72% of the population live in rural areas (*Office of The Registrar General & Census Commissioner, India, 2011*), and there is a high prevalence of HIV infection in antenatal clinics (*National AIDS Control Organisation, 2012*). The HIV epidemic is largely driven by heterosexual transmission and is characterized by poor socio-economic conditions and high levels of illiteracy in the HIV population (*Alvarez-Uria et al., 2012d*). HIV testing is offered free of cost in any of the 71 governmental integrated and counselling testing centres spread across the district, and it is also available in some private clinics. The Rural Development Trust (RDT) is a non-governmental organization that provides medical care to HIV infected people free of cost, including HIV testing, medicines, consultations or hospital admission charges. RDT has three hospitals in the district, and in one of them (Bathalapalli RDT Hospital) CD4 count enumeration and ART are provided free of cost by the Indian Government under a public-private partnership. The Vicente Ferrer HIV Cohort Study (VFHCS) is an open cohort study of all HIV infected patients who have attended RDT hospitals since June 2006. At enrolment in the cohort, the date of the first positive HIV testing is collected. The characteristics of the cohort have been described in detail elsewhere (*Alvarez-Uria et al., 2012d*). The cohort is fairly representative of the HIV population in the district as it covers approximately 70% of all HIV infected patients registered in the district (*Alvarez-Uria et al., 2012b*).

For this study, we selected HIV infected adults (>15 years) living in Anantapur and diagnosed with HIV between January 1st 2007 and November 4th 2011 that were included in the VFHCS. The selection of patients from the database was executed on September 14th

2012. Patients who were lost to follow up were actively searched for by at least one phone call and one home visit by outreach workers and for patients who had died, the date of death was collected.

## Design

To describe the entry into HIV care, we calculated the time period between the HIV diagnosis and the first determination of the CD4 count. Following recommendations based on studies performed in sub-Saharan Africa (*Fox, Larson & Rosen, 2012*), delayed entry into care was defined if patients did not have a CD4 count measurement within three months of HIV diagnosis.

In the analysis, patients' communities were classified as "socially disadvantaged" if the patient self-reported living in a scheduled caste or scheduled tribe community. The "socially disadvantaged" designation was based on the fact that scheduled caste communities are marginalized in the traditional Hindu caste hierarchy and, therefore, suffer social and economic exclusion and disadvantage, and scheduled tribe communities are geographically isolated with limited economic and social contact with the rest of the population. These disadvantaged communities are supported by positive discrimination schemes operated by the Government of India (*Alvarez-Uria, Midde & Naik, 2012a*). Patients were considered to be living near an ART centre if they lived in a mandal with an ART centre or lived next to a mandal (administrative subdivision of districts in Andhra Pradesh; e.g., Anantapur District has 64 mandals) with an ART centre. ART centres are healthcare facilities able to perform CD4 count and provide ART to patients free of cost. During the study period, two mandals had an ART centre. Patients were considered to be living near a town when they lived in any of the six mandals containing a town with a population >100,000 people. Towns have better communications than rural areas. Poverty was defined as living with less than 1000 Indian rupees per month (approximately 18 US dollars in March 2013). In Anantapur, distances between the headquarters of contiguous mandals can range between 10 to 30 km.

## Statistical analysis

Statistical analysis was performed using Stata Statistical Software (Stata Corporation, Release 11, College Station, Texas, USA). We investigated factors associated with entry into care within three months of HIV diagnosis. All independent variables were included in the multivariable analysis. When variables had more than two groups, the group with the largest sample size was selected to avoid collinearity problems (*Katz, 2011*). In order to find adjusted risk ratios that exhibit robustness to outliers, a Poisson regression with robust variance was applied (*Zou, 2004*). All two-way interactions were checked, and we found interactions between gender and other variables. Hence, stratified-by-gender analysis was also performed. Missing values were imputed using multiple imputations by chained equation (*Royston, 2009*). Cumulative incidence of entry into care and death was estimated using competing-risk regression (stcompet command in Stata) (*Cleves, Gould & Gutierrez, 2008*; *Coviello & Boggess, 2004*).

**Table 1** Baseline characteristics and factors associated with entry into care within 3 months of 7701 patients diagnosed with HIV infection from 2007 to 2011 in Anantapur District, Andhra Pradesh, India.

| | Baseline characteristics (n = 7701) N (%) | Entry <3 months N (%) | Multivariable analysis of factors associated with entry into care within 3 months | | |
| --- | --- | --- | --- | --- | --- |
| | | | Overall aRR (95% CI) | Males aRR (95% CI) | Females aRR (95% CI) |
| **Age (years)** | | | | | |
| <25 | 1543 (20.04) | 1021 (66.17) | 0.97 (0.93–1.01) | 0.91* (0.84–0.99) | 1.00 (0.95–1.06) |
| 25–35 | 3362 (43.66) | 2323 (69.1) | 1 (reference) | 1 (reference) | 1 (reference) |
| 35–45 | 1897 (24.63) | 1383 (72.9) | 1.04* (1.00–1.08) | 1.05* (1.01–1.10) | 1.02 (0.96–1.07) |
| >45 | 899 (11.67) | 683 (75.97) | 1.06* (1.02–1.11) | 1.08* (1.02–1.14) | 1.05 (0.98–1.13) |
| Disadvantaged community | 2193 (28.48) | 1466 (66.85) | 0.95* (0.92–0.98) | 0.97 (0.92–1.01) | 0.93* (0.88–0.97) |
| Illiteracy | 4520 (58.75) | 3062 (67.74) | 0.91* (0.88–0.94) | 0.91* (0.87–0.94) | 0.92* (0.88–0.96) |
| **Marital status** | | | | | |
| Divorced/separated | 423 (5.51) | 304 (71.87) | 1.05 (0.99–1.12) | 1.09 (1.00–1.18) | 1.03 (0.95–1.12) |
| Married | 5558 (72.43) | 3840 (69.09) | 1 (reference) | 1 (reference) | 1 (reference) |
| Unmarried | 431 (5.62) | 309 (71.69) | 1.05 (0.98–1.12) | 1.08* (1.00–1.15) | 1.00 (0.82–1.23) |
| Widowed | 1262 (16.45) | 942 (74.64) | 1.08* (1.04–1.13) | 1.00 (0.92–1.08) | 1.12* (1.07–1.17) |
| Homeless | 856 (11.68) | 420 (49.07) | 0.71* (0.67–0.77) | 0.68* (0.61–0.75) | 0.76* (0.69–0.85) |
| Living near an ART centre | 2481 (32.22) | 1789 (72.11) | 1.03 (1.00–1.06) | 1.03 (0.98–1.07) | 1.03 (0.98–1.07) |
| Living near a town | 3316 (43.06) | 2387 (71.98) | 1.03* (1.00–1.06) | 1.01 (0.97–1.06) | 1.05* (1.01–1.10) |
| Poverty | 2740 (36.8) | 1768 (64.53) | 0.96* (0.92–0.99) | 0.93* (0.88–0.97) | 0.99 (0.95–1.05) |
| **Year of HIV diagnosis** | | | | | |
| 2007 | 1715 (22.27) | 941 (54.87) | 1 (reference) | 1 (reference) | 1 (reference) |
| 2008 | 1640 (21.3) | 1057 (64.45) | 1.14* (1.08–1.21) | 1.09* (1.01–1.17) | 1.22* (1.13–1.33) |
| 2009 | 1618 (21.01) | 1140 (70.46) | 1.25* (1.19–1.32) | 1.17* (1.09–1.26) | 1.36* (1.26–1.47) |
| 2010 | 1562 (20.28) | 1277 (81.75) | 1.39* (1.33–1.46) | 1.31* (1.23–1.40) | 1.50* (1.39–1.62) |
| 2011 | 1166 (15.14) | 995 (85.33) | 1.42* (1.35–1.50) | 1.38* (1.29–1.48) | 1.49* (1.38–1.61) |
| **Reason for HIV testing** | | | | | |
| Symptoms | 5006 (65) | 3615 (72.21) | 1 (reference) | 1 (reference) | 1 (reference) |
| Pregnancy | 371 (4.82) | 256 (69) | 0.91* (0.85–0.99) | – | 0.91* (0.84–0.98) |
| HIV+ partner | 1568 (20.36) | 1040 (66.33) | 0.89* (0.85–0.92) | 0.85* (0.79–0.91) | 0.90* (0.86–0.95) |
| Others | 756 (9.82) | 499 (66.01) | 0.95 (0.90–1.00) | 0.97 (0.91–1.04) | 0.91* (0.84–0.99) |
| Female gender | 3397 (44.11) | 2417 (71.15) | 1.08* (1.05–1.12) | | |

**Notes.**

* $P$ value <0.05. ART, antiretroviral therapy; aRR, adjusted risk ratio by Poisson regression with robust variance; CI, confidence interval.

## RESULTS

During the study period, 7701 patients were diagnosed with HIV infection, and 5410 (70.3%, 95% CI, 69.2–71.3) entered into care within three months of HIV diagnosis. Baseline characteristics and factors associated with entry into care within three months overall and by gender are presented in Table 1. 64% of patients were <35 years old and 44% were women. 28% belonged to a disadvantaged community, 59% were illiterate, 72% were married and 37% were living with less than 1000 Indian rupees per month. 43% lived near

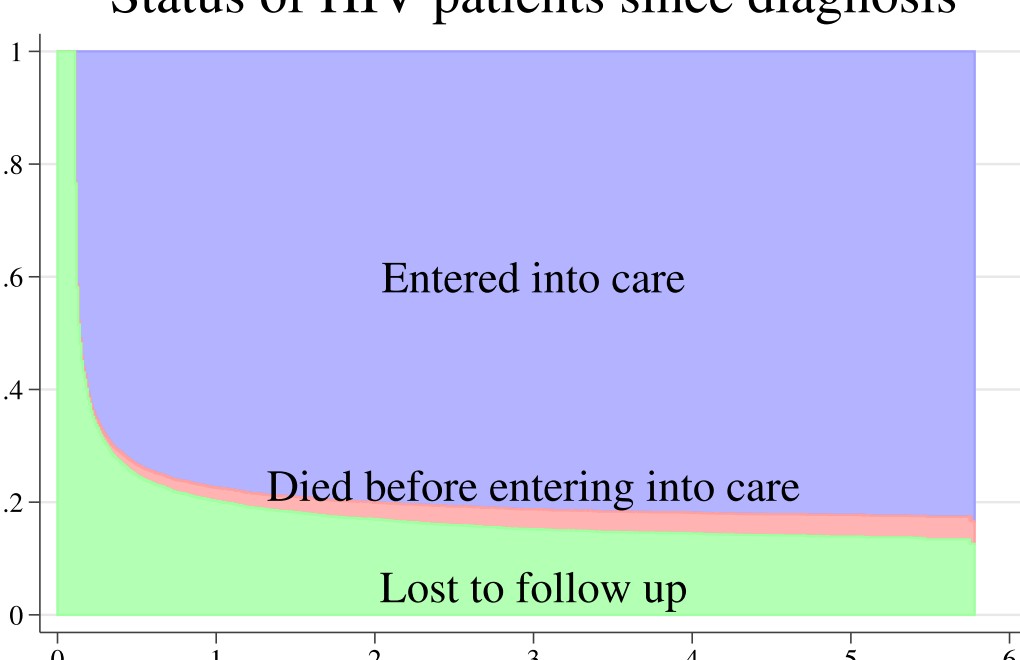

## Status of HIV patients since diagnosis

**Figure 1 Stacked graph of the status of 7701 patients diagnosed with HIV infection from 2007 to 2011 in Anantapur District, Andhra Pradesh, India.** Entry into care was calculated using the time period between the HIV diagnosis and the first CD4 count determination. Patients were considered lost to follow up until they entered into care.

a town and 32% lived near an ART centre. Almost two-thirds of patients were diagnosed because they had symptoms related to HIV.

Factors associated with delayed entry into care were homelessness, and illiteracy. A more recent calendar year of HIV diagnosis, being symptomatic at HIV diagnosis, and female gender were factors associated with entry into care within three months. Younger age, poverty, and being diagnosed with HIV due to having an HIV positive partner were associated with delayed entry into care only in men. Women belonging to disadvantaged communities or living far from a town were less likely to engage in HIV care within three months. Widows and unmarried men were more likely to enter into care within three months than married people.

The estimated cumulative proportion of patients who entered into care after 1, 2, 3, 4 and 5 years since HIV diagnosis was 77.4% (95% CI, 76.5–78.3), 80.1% (95% CI, 79.2–81), 81.2% (95% CI, 80.4–82.1), 81.9% (95% CI, 81–82.7), and 82.3% (95% CI, 81.3–83.1) respectively. A stacked graph of the proportion of patients who entered into care, died before entering into care and were lost to follow up over time is presented in Fig. 1.

## DISCUSSION

This study shows that over two-thirds of patients diagnosed with HIV enter into care within three months and almost 80% within a year. However, most patients who do not

enter into care within a year of HIV diagnosis are lost to follow up. These patients may die before enrolling in an ART centre and are at higher risk of late ART initiation, which has been related to shorter life expectancy, higher cost of medical care and poorer quality of life (*Farnham et al., 2013*; *Johnson et al., 2013*).

These results highlight the need to improve the linkage between HIV testing and ART centres in India. All patients diagnosed with HIV should be monitored until they engage in ART centres (*Thompson et al., 2012*). Both healthcare centres performing HIV testing and ART centres should share responsibility to monitor these patients, and establishing channels of communication is paramount. Rather than passive referrals, healthcare workers from centres where HIV testing is performed should motivate patients diagnosed with HIV by emphasizing the benefits of receiving specialized medical treatment and by offering complete information about the nearest ART centres. In some cases, several counselling sessions may be needed because this strategy has shown positive results in a study conducted in the United States (*Gardner et al., 2005*). In patients who do not reach ART centres, previous studies have shown that outreach visits might be beneficial (*Bradford, Coleman & Cunningham, 2007*; *Naar-King et al., 2007*). However, most of these studies come from developed countries, so further research is needed to investigate new strategies aimed at improving the linkage between HIV diagnosis and HIV treatment in resource-limited settings.

This is one of the first studies to describe predictors of delayed entry into HIV medical care in a resource-limited setting outside sub-Saharan Africa. In accordance with a meta-analysis of studies conducted in sub-Saharan Africa (*Mugglin et al., 2012*), men and patients with low socio-economic status were less likely to enter into care within three months of HIV diagnosis. In a previous study from our cohort, over half of male patients acquired HIV through commercial sex and 21% were not able to reveal their HIV status to their wives (*Alvarez-Uria et al., 2012d*). In the present study, married men were less likely to enter into care within three months than unmarried men, suggesting that some married men preferred to conceal their HIV status rather than to obtain medical care. Homelessness, poverty, illiteracy and belonging to a disadvantaged community were factors associated with delayed entry into care. Illiteracy has been associated with poorer health-related quality of life in HIV patients living in rural India (*Vigneshwaran et al., 2013*). It is possible that for people living in very poor socio-economic conditions, searching for HIV medical care is not the first priority if they are able to cope with their usual daily activities (*Dhillon et al., 2012*).

Similar to a South-African study (*Losina et al., 2010*), patients living far from urban areas and being asymptomatic at the moment of HIV diagnosis were less likely to enter into care, especially women. Patients living in rural areas have more difficulty reaching ART centres and concealing their HIV status than HIV patients living in urban areas, where there are better communications and more anonymity (*Heckman et al., 1998*). Having symptoms at the time of diagnosis may motivate patients to attend ART centres searching for a cure for their symptoms. On the other hand, patients who do not have symptoms

may be more concerned about the stigma and discrimination associated with HIV in India (*Steward et al., 2013*).

The study has some limitations. Although the cohort is fairly representative of the HIV population in the district, we did not have information for all patients diagnosed with HIV in the district, so patients diagnosed in other healthcare facilities who never came to our hospital were not included in the study. Therefore, the proportion of patients diagnosed with HIV that entered into care is likely to be an overestimation. In addition, despite outreach visits and phone calls were performed to re-engage patients lost to follow up, we do not know the proportion of those who enrolled in other ART centres.

## CONCLUSIONS

In our setting, over two-thirds of the patients diagnosed with HIV enter into care within three months and nearly 80% within a year, but most patients who do not enter into care within a year remain lost to follow up. Male gender, homelessness, illiteracy, poverty, living far from a town and being asymptomatic at the moment of HIV diagnosis were factors associated with delayed entry into care after HIV diagnosis. This information could be used to improve the linkage between healthcare centres performing HIV testing and ART centres in India by offering better support to patients within these risk groups.

## ACKNOWLEDGEMENTS

We would like to thank Leann Johnson for her critical review of the manuscript.

### Funding

No external funding was received for this work.

### Competing Interests

The author declares no competing interests.

### Author Contributions

- Gerardo Alvarez-Uria conceived and designed the experiments, performed the experiments, analyzed the data, contributed reagents/materials/analysis tools, wrote the paper.

### Human Ethics

The following information was supplied relating to ethical approvals (i.e. approving body and any reference numbers):

The study was approved by the Ethics Committee of the Rural Development Trust Hospital (Reference number OS/003). Written informed consent was given by patients or caretakers for their information to be stored in the study database and used for research.

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
