# Peer review of "Factors associated with delayed entry into HIV medical care after HIV diagnosis in a resource-limited setting: Data from a cohort study in India"

_PeerJ, doi:10.7717/peerj.90_

## Round 0.1 · original submission · Minor Revisions

Dear Dr. Alvarez--‐Uria,

Despite all my efforts to find another reviewer for your paper entitled "Factors associated with delayed entry into HIV medical care after HIV diagnosis in a resource--‐limited setting: data from a cohort study in India", I have not succeeded. However, as I am sufficiently expert in this field to submit a review, I have acted as the second reviewer on your submission. Please find my review below.

Considering my review and the review of an expert in this field, my overall decision is that your paper should be accepted for publication at PeerJ, once the minor issues raised by the review process are properly addressed.

Thanks a lot for considering PeerJ for publishing your work.

With best regards Dr. Tomas Perez--‐Acle Editor PeerJ

Review:

This work is primarily intended to investigate the risk factors associated with a delayed entry into patient--‐care measurements in a large cohort study in India, comprising 7701 HIV--‐positive patients. According to the results, substantial gender and income--‐related differences were found. As seen from the references, this study is presented as a natural consequence of a previous one (Alvarez--‐Uria, 2012c).

Despite that the overall performance of this study could fit the standards of PeerJ, several issues should be properly addressed prior to its publication in PeerJ. Please address the following issues:

Introduction.

lines 11--‐13: Does the author statement imply that ARTs can only play a role in low and middle--‐income countries? How this role can be compared to that of ARTs in the developed world? To improve the validity of this paragraph, the author should modify it to include a reference to the role of ARTs in the developed world.

line 14: Please improve the connection between this paragraph and the previous one. Does the author imply that the link between diagnosed people and facilities providing ART occur only in low and middle--‐income countries? What happens in the developed world?

line 16: In accordance to previous comments, the works of Adler and Althoff, demonstrate that late enter to care after HIV diagnosis is not only a matter of low--‐ income countries. This paragraph should be better linked to previous ones.

Methods.

Statistical Analysis: the whole set of statistical analyses that were performed is not sufficiently justified neither explained. Despite that these methods were applied elsewhere, a justification for each applied method should be used. For instance, line 65 should be read “In order to find adjusted risk ratios that exhibit robustness to outliers, a Poisson regression with robust variance was applied”. On the other hand, the author selected a set of methods designed to compare pair--‐wise variables. Up to what extent the findings of this paper could be valid when using tuples of higher order? A PCA analysis should be interested to conduct to determine how much variability is associated to each variable.

Reviewer 1 ·

Basic reporting

The author succeeded in conforming to journal standards with respect to basic reporting, with some minor exceptions, detailed below:
1. In the introduction, lines 11-12, the authors start out with a statement about low-and middle income countries "Despite the impressive roll-out of antiretroviral therapy (ART) in low and middle-income
countries," and connect these ART programs to outcomes not just in low- and middle-income countries, but globally: "1.7 million people died of HIV-related pathologies in 2011 (UNAIDS, 2012). The author then goes on to reference studies in developed, high-income countries to support the statement in lines 16-17 "However, the majority of patients enter care late..." The author should revise lines 11-12 to say
"Despite the impressive roll-out of antiretroviral therapy (ART) programs worldwide, including in low and middle-income countries...." to improve the logical flow of ideas in this section.
2. A more detailed description of the context of the study would be helpful. In particular, if such figures are available, it would be helpful to know what percentage of the population in Anantapur lives in rural areas, how many towns of population size >100,000 people and how many RDT hospitals and ART centres are located in Anantapur, and also how many of the 64 mandals had an ART centre and how many had an RDT hospital. It would also be helpful to know something about HIV testing in the district--is it free, where is it offered, and to whom? Finally, how did patients get included in the cohort from which the study population was drawn? Were they from all RDT hospitals in the district or a limited set?
3. In line 110, please consider changing the word "strengthening" to "emphasizing" and "facilitating" to "offering."

Experimental design

The research question and methods are, in general, described well. However, some minor rewording and some additional information would be helpful, specifically:
1. In line 37-38, please given a brief description of the cohort in addition to citing previously published characteristics. The reader should not have to look up another paper to understand basic information such as the inclusion and exclusion criteria for the cohort from which the study population was drawn, including, if possible, how many RDT hospitals contributed patients to the cohort, and the reasons for the visits to the RDT hospitals that led to enrollment in the cohort--did some patients attend the hospital to get tested for HIV, were all patients tested routinely at the hospital or because they wished to be tested or because they were sick? It would be helpful for the reader to have some sense of how representative the patients in the cohort are of all those with HIV infection in the district. Are there are other clinical facilities that HIV patients might go that might preclude their attendance at an RDT hospital?
2. In line 49, the sentence, "Designation of the community of patients was performed by self-identification" is unclear. Please explain the designation of community employed in the analysis, i.e., "In the analysis, patients' communities were classified as "socially disadvantaged" if the patient self-reported living in a scheduled caste or scheduled tribe community. The "socially disadvantaged" designation was based on the fact that scheduled caste communities are marginalized in the traditional Hindu caste hierarchy and, therefore, suffer social and economic exclusion and disadvantage, and scheduled tribe communities are geographically isolated with limited economic and social contact with the rest of the population."
3. In lines 64-66, please define what is meant by "near an ART centre" and "near a town" in terms of ranges of distance, if possible.
4. In the statistical analysis section starting on line 63, the description of the Poisson regression lacks important details. Please explicitly state the independent and dependent variables included and what criteria were used to decide what to include in the model and to decide which levels or categories of the independent variables were selected as the reference. How was interaction evaluated?
5. Please describe the calls and outreach--how many attempts were made before the patient was determined to be "lost to follow-up?."

Validity of the findings

Below are some errors of commission and omission that should be corrected.
1. The rationale for choice of reference categories is unclear and is likely to have affected the results. For example, the oldest or youngest group seems a more intuitive choice for the reference category for age. No mention is made in the narrative that men in the older age groups were more likely to enter care in the first 3 months after diagnosis--perhaps this finding should be pointed out?
2. In line 86-87, the association of social disadvantage with delayed care entry is presented as if it holds for the whole study population, but this association was observed for women only.
3. In lines 87-88, which factor is associated with the aRR provided--female gender? The others aren't listed in the table. Did the author intend to provide aRRs for all the factors listed? What specifically is meant by "later calendar year?"
4. In line 93, please clarify that these are estimated cumulative proportions.
5. Please describe how many of the patients who appeared to be lost to follow up were eventually dispositioned in another category as a result of information gained from calls or outreach.
6. In the limitations section starting on line 140, please discuss how likely the study population is to be representative of all HIV-infected persons in Anantapur. Is it likely that a significant number of HIV-infected individuals in the district are undiagnosed? If so, please discuss the implications for the estimation of delayed care entry.
7. Please consider discussing what your data tell you, if anything, about the consequences of delayed care entry to HIV+ individuals and those at risk of acquiring HIV infection in Anantapur.
8. The title of the Table should include the number of patients in the study, that they were diagnosed with HIV from 2007-2011, and that they were diagnosed in Anantapur District, Andhra Pradesh, India. In the Table, please include, if possible, some information about the distribution of patients geographically, by mandal or RDT. In a footnote to the table, please describe the factors included in the model to yield the adjusted risk ratios.
9. The title of the Figure should include the number of patients in the study, that they were diagnosed with HIV from 2007-2011, and that they were diagnosed in Anantapur District, Andhra Pradesh, India. In a footnote to the figure, please include the definitions of entry to care and lost to follow-up.

Additional comments

The author is to be commended for a well-written paper. The paper should be accepted for publication, pending addition of some critical information, and some minor corrections.

---

## Round 0.2 · accepted · Accept

I'd like to thank to the author by addressing all the reviewers comments. Despite that the overall findings of this paper will certainly be of interest to the PeerJ community, we encourage to the author for the inclusion of additional statistical methods, such as PCA and LOOCV, for future contributions in this field.